# Effects of Concurrent Exposure to Chronic Restraint-Induced Stress and Total-Body Iron Ion Radiation on Induction of Kidney Injury in Mice

**DOI:** 10.3390/ijms23094866

**Published:** 2022-04-27

**Authors:** Duling Xu, Hongyan Li, Takanori Katsube, Guomin Huang, Jiadi Liu, Bing Wang, Hong Zhang

**Affiliations:** 1Department of Medical Physics, Institute of Modern Physics, Chinese Academy of Sciences, Lanzhou 730000, China; xuduling21@mails.ucas.ac.cn (D.X.); lihy@impcas.ac.cn (H.L.); huangguomin@impcas.ac.cn (G.H.); liujiadi21@mails.ucas.ac.cn (J.L.); 2Key Laboratory of Heavy Ion Radiation Biology and Medicine, Chinese Academy of Sciences, Lanzhou 730000, China; 3Key Laboratory of Basic Research on Heavy Ion Radiation Application in Medicine, Lanzhou 730000, China; 4School of Nuclear Science and Technology, University of Chinese Academy of Sciences, Beijing 101408, China; 5Advanced Energy Science and Technology Guangdong Laboratory, Huizhou 516029, China; 6National Institute of Radiological Sciences, National Institutes for Quantum Science and Technology, Chiba 263-8555, Japan; katsube.takanori@qst.go.jp

**Keywords:** chronic restraint, psychology stress, ionizing radiation

## Abstract

Concurrent exposure to ionizing radiation (IR) and psychological stress (PS) may affect the development of adverse health consequences in scenarios such as space missions, radiotherapy and nuclear accidents. IR can induce DNA damage and cell apoptosis in the kidneys, thus potentially leading to renal fibrosis, which is the ultimate outcome of various chronic progressive nephropathies and the morphological manifestation of a continuous coordinated response after renal injury. However, little is known regarding the effects of concurrent IR exposure and PS on renal damage, particularly renal fibrosis. In this study, using a chronic restraint-induced PS (CRIPS) model, we exposed *Trp53*-heterozygous mice to total body irradiation with 0.1 or 2 Gy ^56^Fe ions on the eighth day of 28 consecutive days of a restraint regimen. At the end of the restraint period, the kidneys were collected. The histopathological changes and the degree of kidney fibrosis were assessed with H&E and Masson staining, respectively. Fibronectin (FN) and alpha smooth muscle actin (α-SMA), biomarkers of fibrosis, were detected by immunohistochemistry. Analysis of 8-hydroxy-2 deoxyguanosine (8-OHdG), a biomarker of oxidative DNA damage, was performed with immunofluorescence, and terminal deoxynucleotidyl transferase-mediated nick end labeling assays were used to detect apoptotic cells. Histopathological observations did not indicate significant structural damage induced by IR or CRIPS + IR. Western blotting revealed that the expression of α-SMA was much higher in the CRIPS + IR groups than the CRIPS groups. However, no differences in the average optical density per area were observed for FN, α-SMA and 8-OHdG between the IR and CRIPS + IR groups. No difference in the induction of apoptosis was observed between the IR and CRIPS + IR groups. These results suggested that exposure to IR (0.1 and 2 Gy ^56^Fe ions), 28 consecutive days of CRIPS or both did not cause renal fibrosis. Thus, CRIPS did not alter the IR-induced effects on renal damage in Trp53-heterozygous mice in our experimental setup.

## 1. Introduction

A variety of psychological stress (PS) suppresses the immune system [1] and has gradually become the main risk factor of several physiological disorders, which might result in diabetes, metabolic syndrome [2] and the diseases of central nervous and cardiovascular systems, and even cancer. With the rapid development of space technology, an increasing number of astronauts have to suffer from the harmful exposure to cosmic radiation (CR) [3]. CR affects astronaut health, especially the high atomic number and high-energy (HZE) particles that are often referred to as high linear energy transfer (LET) radiation or densely ionizing [4]. High LET ion exposure can lead to damage that is more difficult to repair than X-ray and γ-ray damage [5]. Among the particles of CR, 13% of the dose during manned long-term deep-space Mars missions was from Fe particles alone [6], thus, Fe particles are of great research interest in CR. Inevitably, astronauts are exposed to LET ion radiation [7], microgravity and an enclosed environment during long-term space missions [8]. The exposure to an enclosed environment in space flights might lead to PS. In reality, astronauts are exposed to the dual harms of PS and ionizing radiation (IR) exposure on space missions, and the PS could exacerbate the detrimental effects induced by IR that could result in an additive or even synergistic effect. However, little is known about the combined health consequences of exposure to PS and IR. Therefore, assessing the possible harmful effects of IR and PS and investigating whether PS might modify IR-induced toxicity are important for understanding the possible effects of these exposures on human health.

Chronic restraint-induced PS (CRIPS) models have been successfully used in previous studies, and results show that CRIPS could enhance the frequency of chromosomal exchange induced by ^56^Fe ion irradiation in *Trp53*-heterozygous mice [8]. Although CRIPS has no additive effects beyond the radiation-induced detrimental effects on the hematopoietic system in *Trp53*-wildtype mice [9] and spermatogenic cells in *Trp53*-heterozygous mice [10], the effects from concurrent exposure to CRIPS and IR on other organs and systems are still unclear. Since the kidney is an important organ that regulates body fluid, electrolyte and acid-base metabolism, kidney function and water and electrolyte balance are crucially important in space flights [11]. In the present study, effects from concurrent exposure to CRIPS and IR on kidney were investigated in *Trp53*-heterozygous mice. According to the documented data, CRIPS increases the aldosterone levels that were related with metabolic syndrome and renal injury [12]. Moreover, renal fibrosis is a common outcome of many chronic kidney diseases, independently of the underlying etiology [13]. In chronic renal injury, a fibrous matrix is continually deposited, thus destroying the tissue structure, decreasing the blood supply, interfering with organ function and, finally, leading to renal failure [14]. Chronic toxicity induced by radiation in the kidneys is generally characterized by fibrogenesis and extracellular matrix deposition [15]. However, whether PS is a modifying factor in IR-induced renal injury remains unknown. Thus, we investigated whether PS exacerbates the renal fibrosis induced by ^56^Fe ion irradiation, to better understand whether CRIPS is an additive factor in renal fibrosis induced by IR in *Trp53*^+/−^ mice.

High LET ions cause direct damage by destroying chemical bonds, and also induce indirect damage through the production of reactive oxygen species (ROS) [15]. When ROS exceeds the antioxidant capacity, oxidative stress and DNA damage increase [16]. After radiation injury, renal dysfunction due to DNA damage leads to cell apoptosis and, eventually, renal fibrosis [17]; moreover, the cost of treatment is expensive. These factors interact with mental stress, thus affecting the progression and prognosis of renal fibrosis [18]. Here, we used ^56^Fe ion radiation to induce DNA damage in mice to explore whether CRIPS might alter the effects of IR on renal fibrosis. We used a chronic restraint model to simulate CRIPS and used an ^56^Fe ion beam to deliver total-body irradiation (TBI) with 0.1 and 2 Gy in *Tp53*-heterozygous male mice. Fibronectin (FN) [19] and alpha smooth muscle actin (α-SMA) [20], biomarkers of fibrosis, were detected by immunohistochemistry. The expression of α-SMA was detected by western blotting, and the levels of 8-hydroxy-2 deoxyguanosine (8-OHdG) were used to assess DNA damage [21] by immunofluorescence. We hypothesized that IR causes DNA damage, thus inducing apoptosis and enhancing renal fibrosis, and that CRIPS and concurrent exposure to IR might have additive effects on renal fibrosis.

## 2. Materials and Methods

### 2.1. Animal Treatment

The animal experiments were conducted at the National Institute of Radiological Science (NIRS) of Japan. Four-week-old C57BL/6N TP53 heterozygous (*Trp53*^+/−^) mice (BRC NO. 01361) were used in the experiments. From 07:00 to 19:00, we placed one or two mice in an autoclaved cage maintained under controlled temperature (23 ± 2 °C) and humidity (50 ± 10%) under 12 h continuous light/dark cycle conditions, and allowed them to freely consume standard laboratory chow (MB-1; Funabashi Farm Co., Funabashi, Chiba, Japan) and acidified water (pH 3.0 ± 0.2). One week before the experiment, the mice were randomly divided into six groups (Table 1): a control group (*n* = 6), which did not receive restraint and ^56^Fe ion radiation; a 0.1 Gy ^56^Fe ion radiation group (*n* = 6); a 2 Gy ^56^Fe ion radiation group (*n* = 6); a CRIPS group (*n* = 6), receiving only chronic restraint; a CRIPS + 0.1 Gy ^56^Fe ion radiation group (*n* = 6); and a CRIPS + 2 Gy ^56^Fe ion radiation group (*n* = 6). In the restraint group, a flat-bottomed rodent holder (RSTR541; Kent Scientific Co., United States) placed horizontally in the cage was used for chronic restraint for 28 days for 6 h per day (09:30–15:30). The mice were then placed in the same cage for free activity (15:30–09:30), during which the mice were allowed to eat and drink freely. In the early morning (3:30–04:30 a.m. or 6:00–7:00 a.m.), on day 8 of the 28-day restraint regimen, ^56^Fe ions of TBI were irradiated at NIRS with a heavy ion medical accelerator in Chiba (HIMAC) at doses of 0.085 Gy/min, 1.1–2.7 Gy/min and 0.1 and 2 Gy (500 MeV/nucleon, 200 keV/µm).

### 2.2. The Histological Observation

The mice were anesthetized by inhalation of carbon dioxide and then euthanized by cervical dislocation. The kidneys were separated and fixed in 4% paraformaldehyde (Solarbio Life Sciences, Beijing, China), then embedded in paraffin and cut into sections with 4 µM thickness. The sections were stained with hematoxylin and eosin (H&E) (Solarbio) after paraffinization and rehydration, then mounted with neutral gum [22]. 

### 2.3. Masson Staining

The Masson staining method was used to detect collagen fibers in the kidneys. Paraffin embedded renal tissue sections were used for dewaxing and hydration. Hematoxylin was used to stain nuclei for 5 min and was washed away with Tris Buffered Saline (TBS) for 10 min. Subsequently, Masson solution staining (Servicebio Technology Co., Wuhan, Hubei province, China) was performed for 5 min. The sections were then placed in 1% glacial acetic acid solution (Servicebio), differentiated for 5–10 s, then mounted with neutral gum. Images were taken under an optical microscope. Four fields were randomly selected from each section for analysis [23].

### 2.4. TUNEL Assay

Detection of apoptosis was performed with a TdT-mediated dUTP nick end labeling (TUNEL) assay kit (Servicebio). The sections were deparaffinized and hydrated, then permeabilized with proteinase K at 37 °C for 15 min and treated with 3% H_2_O_2_ for 15 min. After being washed with 0.01 M phosphate-buffered saline, the sections were incubated with TUNEL reaction mixture (1:5:50 ratio of TDT enzyme, dUTP and buffer) for 1 h at 37 °C in a dark humidified plastic container. Reagent Streptavidin-Horseradish Peroxidase (HRP) (Servicebio) and Tris Buffered Saline with Tween (TBST) were mixed at a ratio of 1:200, added to the sections and incubated in a 37 °C incubator for 30 min. Then, sections were dried slightly, and freshly prepared 3, 3′-diaminobenzidine (DAB) (Servicebio) chromogenic reagent was added to marked sections. Finally, the sections were stained with hematoxylin after paraffinization and rehydration, then mounted with neutral gum [24]. 

### 2.5. Immunofluorescence

The sections were deparaffinized and hydrated, then immersed in 3% H_2_O_2_ for 30 min, then incubated in 1% Triton X-100 for 30 min, blocked for 25 min at room temperature in 5% bovine Serum Albumin (BSA) (Solarbio) (5 g BSA in 100 mL TBS) and incubated in primary antibody in TBS at 4 °C. Sections were stained with 5 g/mL 4′, 6-diamidino-2-phenylindole (DAPI) (Soliabo) for 10 min, washed with TBS twice for 5 min, then mounted with glycerin–sodium bicarbonate wet sealant [25]. 

### 2.6. Immunohistochemistry

Immunohistochemistry was used to detect the expression of fibronectin (ab268020) and α-SMA (ab124964) (Abcam, Cambridge, UK). The renal sections were deparaffinized and hydrated, then maintained at 96 °C for 12 min for antigen repair. After being treated with 3% hydrogen peroxide for 15 min, the primary antibody (1:500) was incubated with the sections at 4 °C overnight, and then a Streptavidin-Peroxidase kit (Bioss Biotechnology Co., Beijing, China) was used for immunoreaction. Bound peroxidase activity was visualized with a DAB detection kit (Bioss). Sections were then counterstained with hematoxylin and mounted [26]. 

### 2.7. Western Blotting

Two samples were randomly selected as one independent sample for protein extraction per group. The total protein of renal tissue from two samples was extracted with RIPA buffer (Solarbio), and the protein concentration was measured with the BCA Kit (Solarbio). A total of 40 μg of protein was separated on 10% sodium dodecyl sulfate polyacrylamide gel electrophoresis (SDS-PAGE), then transferred to 0.45 µM polyvinylidene fluoride (PVDF) membranes (Millipore, Billerica, MA, USA). The membranes were blocked with 5% skimmed milk for 1 h at room temperature, then washed with TBST. Primary antibody (α-SMA, 1:2000, Servicebio) was incubated at 4 °C overnight. After secondary antibody binding, a chemiluminescent reagent kit (New Cell and Molecular Biotech, Suzhou, China) was used to detect protein bands. β-actin was used as a control, and the intensity of protein bands was analyzed in Amersham Imager 680 software (GE Healthcare Bio-Sciences AB, Uppsala, Sweden).

### 2.8. Statistical Analysis

The sections were scanned with Panoramic MIDI software (3DHISTECH, Budapest, Hungary). Image-Pro Plus software (Media Cybernetics, Inc, Bethesda, MD, USA) was also used to analyze and the average optical density (AOD) (Integrated optical density/area) to indicate the level of protein expression. Five different visual immunohistochemistry fields were taken from each sample (200×), and the average AOD was calculated. The statistical data for each group were analyzed in Statistical Package for Social Sciences (SPSS) 23.0 (IBM Corp., Armonk, NY, USA), and the statistical data are expressed as mean ± standard deviation. The means across multiple groups were analyzed with one-way analysis of variance (ANOVA), and the means between two groups were analyzed with *t*-test, with *p* < 0.05 indicating significance.

## 3. Results

### 3.1. The Histopathological Observation

H&E staining was used to compare the renal structural changes between the control group and the experimental groups (Figure 1). The cortical glomeruli were normal, and the basement membrane was clear in the control and CRIPS groups. Compared with the control group, the IR and CRIPS + IR groups had smaller volumes of glomeruli, and the differences were more pronounced in the 2 Gy and 2 Gy + IR groups, in which renal interstitial inflammatory cell infiltration was also observed. 

### 3.2. The 8-OHdG Expression

As a marker of DNA damage, 8-OHdG is highly reliable and abundant. Thus, we assessed 8-OHdG expression with immunofluorescence. The 8-OHdG fluorescence was detected almost exclusively in the cell nuclei on the surface of the renal cortex in the control and CRIPS groups. The fluorescence signal was greater in the eosinophils or basophils on the surface of the renal cortex in the IR and CRIPS + IR groups than the control group (Figure 2A). However, no significant differences in fluorescence intensity were observed among groups (Figure 2B).

### 3.3. The Apoptotic Cells Detection

Apoptotic renal cells were detected with TUNEL assays. Eosinophils or basophils were observed in the control and CRIPS groups, and no clear differences in apoptotic cell types were observed between the IR and CRIPS + IR groups, including the 2 Gy and CRIPS + 2 Gy groups and the control group (Figure 3).

### 3.4. The Evaluation of Renal Fibrosis 

Renal fibrosis was evaluated with Masson staining. The collagen fibers in the glomeruli and renal tubules showed weak blue staining in the six groups (Figure 4A). The basement membrane was stained blue in only the control and CRIPS groups. The degree of staining showed no clear differences between the 0.1 Gy or CRIPS + 0.1 Gy groups and the control group. The area of collagen fibers in the 2 Gy and 2 Gy + IR groups was greater than that in the 0.1 Gy and CRIPS + 0.1 Gy groups. However, no differences were observed among all groups after statistical analysis (Figure 4B).

### 3.5. The FN and α-SMA Expression

FN and α-SMA are important biomarkers of fibrosis [18]. Thus, the expression and distribution of FN and α-SMA were detected by immunohistochemistry. FN was expressed in the renal tubular basement membrane, adventitia and glomerular mesangium, but the staining was weak in the control and CRIPS groups (Figure 5A). The FN staining was mainly located in the glomeruli, mesangium and renal tubular basement membrane in the IR and CRIPS + IR groups (Figure 5A). However, no significant differences were observed among all groups (Figure 5B). α-SMA was expressed in arterial middle smooth muscle cells but not renal stroma in the control and CRIPS groups (Figure 6A). The expression of α-SMA was elevated in renal tubules and the renal interstitium in the IR and CRIPS + IR groups, and no difference in the AOD of α-SMA expression was observed among groups (Figure 6B). However, the expression of α-SMA was higher in the CRIPS + IR groups than the CRIPS groups (Figure 7B). 

## 4. Discussion

Radiation-induced fibrosis is a late sequela of both therapeutic and accidental irradiations, which has been described in various tissues, including the lung, liver, kidney and skin [27]. The altered renal lipid metabolism induced by chronic restraint stress may contribute to renal injury [12]. Thus, the harmful effects from concurrent exposure to IR and PS on renal injury and fibrosis cannot be ignored during long-term space missions. The present study explored the effects of a low dose of ^56^Fe ion radiation combined with CRIPS on renal fibrosis, and whether CRIPS might be an additive factor in the renal fibrosis induced by IR in *Trp53*^+/−^ mice.

Radiation nephritis (RN) is a clinical syndrome caused by acute or chronic kidney injury after IR. The effects in patients include fatigue, shortness of breath, edema and significant renal failure [3]. Chronic RN may be caused by the long-term evolution and aggravation of acute RN, including renal fibrosis [28]. In this study, we used a chronic restraint experimental device to generate a model of *Trp53*^+/−^ male mice exposed to ^56^Fe radiation under PS, in which we observed the renal fibrosis. At the end of the 28-day experimental period, we conducted histochemical experiments on the renal tissue, which indicated no structural differences between the control group and the experimental groups. FN and α-SMA staining revealed no significant differences in fibrosis between the control group and the experimental groups, including the 2 Gy and CRIPS + 2 Gy groups. The above results indicated that CRIPS, IR and CRIPS + IR did not damage the renal structure or induce the renal fibrosis. 

The early biological response to high LET radiation exposure, namely the induction of ROS, has been demonstrated in previous studies, suggesting that ROS and nitrogen species formed after IR might trigger DNA damage and lead to localized inflammation [29]. This inflammatory process ultimately evolves into a fibrotic process characterized by increased collagen deposition, poor vascularity and scarring [29]. Previous work has indicated that CR causes oxidative damage by increasing ROS in the kidneys [30]. Therefore, in this study, we explored the process through which IR-induced DNA damage leads to renal cell apoptosis, thus accelerating renal fibrosis. We examined the renal cell distribution of the oxidative DNA damage marker protein 8-OHdG (Figure 2A), which was present in the nucleus, thus indicating that oxidative DNA damage destroyed the nucleoli of renal cells. However, no significant differences in the AOD of 8-OHdG were observed between the control group and the experimental groups, including the 2 Gy and CRIPS + 2 Gy groups. TUNEL assays also indicated few apoptotic cells in the IR and CRIPS + IR groups, including the 2 Gy and CRIPS + 2 Gy groups. These results indicated that CRIPS, IR and CRIPS + IR did not induce DNA damage and cell apoptosis. 

Fe particles, as one of the important components of HZE particles in space, are gradually attracting researchers’ interest [31]. Similar to other heavy ions, ^56^Fe ions also cause damage that is more difficult to repair than conventional radiation-induced damage from X-rays and γ-rays. Turker et al. (2017) explored the effects of ^56^Fe ion exposures on *Aprt* mutant frequency and toxicity in the kidney epithelium. They found that significantly increased levels of apparent mitotic recombination events were observed for the 0.5, 1.0 and 2.0 Gy doses, and a significant reduction in clone of kidney epithelial cells was observed at 1.0 and 2.0 Gy groups [32]. In the present study, we used 0.1 and 2 Gy TBI to simulate the space radiation; no significant differences in histopathological and fibrotic changes, DNA damage and renal cell apoptosis were observed between the IR and CRIPS + IR groups, for two possible reasons. Radiation dose is a major factor in the induction of RN, and the dose used in the present study was not sufficient to cause renal fibrosis. Larger doses appear to be more efficient for IR-induced renal fibrosis. Second, studies on radiation damage in animal experiments have shown more severe damage over time. A previous study indicated injury to renal tubules occurring 3 months after radiation [33], but observed only minor changes in glomeruli. Rats have been found to show proteinuria within 6 weeks after irradiation and uremic morbidity after 26 weeks [27]. Less positive staining of 8-OHdG has been observed in severely atrophic tubules at 24 weeks after irradiation [34]. Our findings are in agreement with the above observations, indicating less positive immunofluorescence staining for 8-OHdG in the nuclei of cells in the 2 Gy and CRIPS + 2 Gy groups. Although the fluorescence intensity of 8-OHdG did not significantly differ among experimental groups, the fluorescence signal of 8-OHdG was enhanced in these groups compared with the control, thus indicating persistent oxidative stress in the kidneys after irradiation. Therefore, observation time is another important factor for studying renal fibrosis. In addition, recovery may occur; thereby, continuous observation over a longer duration after exposure is required.

It should be noted that, for flight crew in a manned long-duration deep-space mission, radiation exposure is chronic and at very low fluences and fluence rates. However, it is very hard, technically, to simulate the exposure in space in experimental studies using heavy ion accelerators. In further studies, it would be practical to use fractionated exposure in combination with longer restraint time to simulate the exposure in space. On the other hand, to verify whether PS could modify radiation-induced renal fibrosis, a larger dose and a longer study duration should be applied to induce renal fibrosis in the experimental model. This is because, clinically, a dose at 23 Gy caused chronic kidney disease in 5% of cases [35], and patients who received total doses of 18.8 Gy with fractional irradiation showed no change in lipid peroxidation or protein oxidation in urine 42 days after radiation [36].

## 5. Conclusions

The histopathological observations revealed no structural changes, and immunohistochemical analysis showed no significant differences in FN and α-SMA expression between the CRIPS + IR and IR groups (compared at the same ^56^Fe ion dose). Although exposure to IR (0.1 and 2 Gy ^56^Fe ions), 28 consecutive days of CRIPS or both did not cause renal fibrosis, these results provide a reference for exploring the renal fibrosis induced by CRIPS, IR and CRIPS + IR. Further studies should be performed to verify the effects of concurrent exposure to both PS and IR under different experimental conditions, such as using lager doses of IR, chronic or fractioned exposures and a longer-term CRIPS model. 

## Figures and Tables

**Figure 1 ijms-23-04866-f001:**
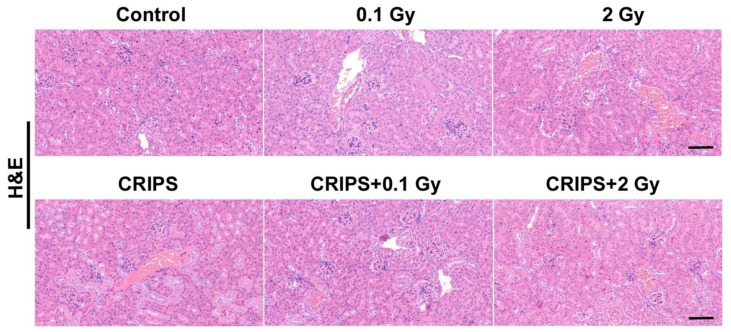
H&E staining of renal sections from *Trp53*^+/−^ mice (magnification, 200×; scale bar = 100 µM). CRIPS, chronic restraint-induced stress.

**Figure 2 ijms-23-04866-f002:**
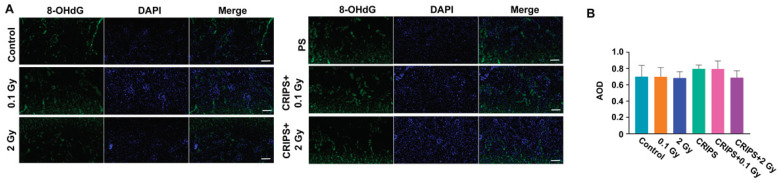
Expression of 8-OHdG, detected in renal sections in *Trp53*^+/−^ mice. (**A**), the expression of 8-OHdG was analyzed by immunofluorescence (magnification, 200×; scale bar = 100 µM); (**B**), the average AOD of 8-OHdG in each group. CRIPS, chronic restraint-induced stress; AOD, the average optical density (Integrated optical density/area).

**Figure 3 ijms-23-04866-f003:**
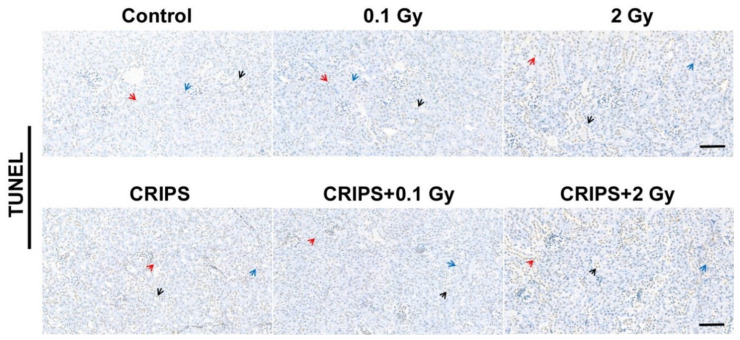
The apoptotic cells of renal sections detected by TUNEL assay in *Trp53*^+/−^ mouse (magnification, 200×; scale bar = 100 µM). Arrows indicate apoptotic cells (red arrow indicates renal tubular epithelial cells; black arrow indicates eosinophils; blue arrow indicates basophils). CRIPS, chronic restraint-induced stress.

**Figure 4 ijms-23-04866-f004:**
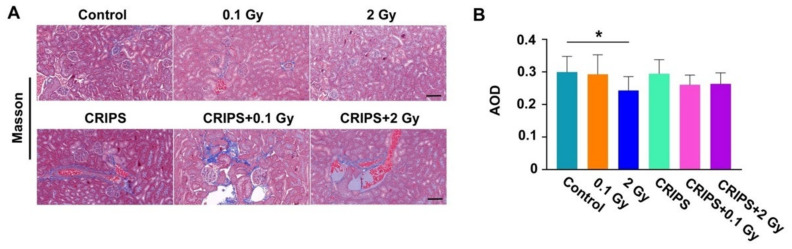
Masson staining of renal sections in *Trp53*^+/−^ mice. (**A**), the blue staining indicates collagen fibers (magnification, 200×; scale bar = 100 µM); (**B**), the average AOD of blue staining in each group. CRIPS, chronic restraint-induced stress; AOD, the average optical density (Integrated optical density/area). * *p* < 0.05.

**Figure 5 ijms-23-04866-f005:**
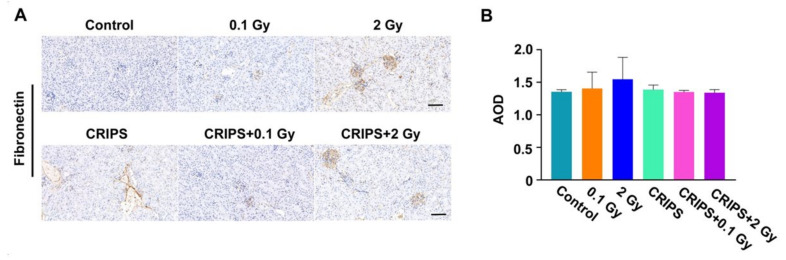
Expression of FN, detected in renal sections in *Trp53*^+/−^ mice. (**A**), the expression of FN was analyzed by immunohistochemistry (magnification, 200×; scale bar = 100 µM); (**B**), the average AOD of FN in each group. CRIPS, chronic restraint-induced stress; FN, Fibronectin; AOD, the average optical density (Integrated optical density/area).

**Figure 6 ijms-23-04866-f006:**
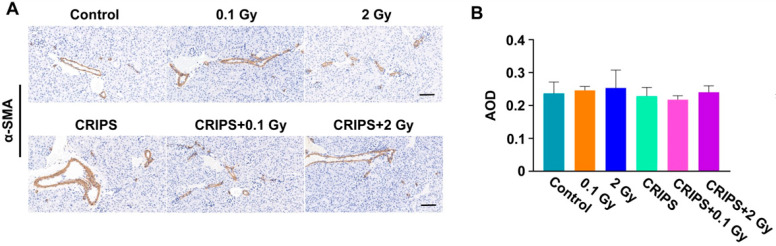
Expression of α-SMA, detected in renal sections in *Trp53*^+/−^ mice. (**A**), the expression of α-SMA was analyzed by immunohistochemistry (magnification, 200×; scale bar = 100 µM); (**B**), the average AOD of α-SMA in each group. CRIPS, chronic restraint-induced stress; AOD, the average optical density (Integrated optical density/area).

**Figure 7 ijms-23-04866-f007:**
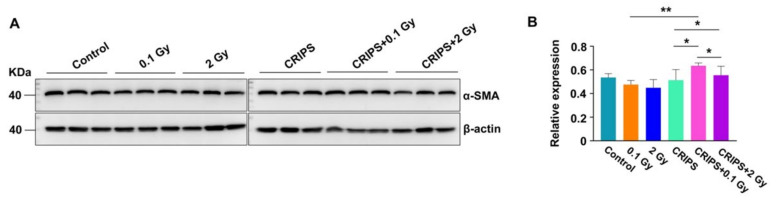
Expression of α-SMA, detected by Western blotting in *Trp53*^+/−^ mouse kidneys. (**A**), the representative images of Western blotting, three replicates in each group; (**B**), the relative expression of α-SMA in each group. CRIPS, chronic restraint-induced stress. * *p* < 0.05, ** *p* < 0.01.

**Table 1 ijms-23-04866-t001:** Experimental group.

Group	Treatment
Control (*n* = 6)	Receiving neither chronic restraint nor TBI with ^56^Fe irradiation
0.1 Gy (*n* = 6)	Receiving only ^56^Fe-TBI at 0.1 Gy
2.0 Gy (*n* = 6)	Receiving only ^56^Fe-TBI at 2.0 Gy
CRIPS (*n* = 6)	Receiving only chronic restraint to simulate chronic restraint-induced psychological stress
CRIPS + 0.1 Gy (*n* = 6)	Receiving 0.1 Gy ^56^Fe-TBI on day 8 of the 28-day restraint regimen
CRIPS + 2.0 Gy (*n* = 6)	Receiving 2.0 Gy ^56^Fe-TBI on day 8 of the 28-day restraint regimen

TBI, total-body irradiation; CRIPS, chronic restraint-induced psychological stress.

## Data Availability

The authors confirm that the data supporting the findings of this study are available within the article.

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
