# Peer review of "Effects of Concurrent Exposure to Chronic Restraint-Induced Stress and Total-Body Iron Ion Radiation on Induction of Kidney Injury in Mice"

_ijms, 2022, doi:10.3390/ijms23094866_

Round 1

Reviewer 1 Report

The study deals with probing the effect of concurrent exposure to ionizing radiation (IR) and chronic restraint-induced stress (CRIPS) through the renal damage in Trp53-heterozygous mice.

The methods used in the study are clear and the results are described in detail. As the main conclusion, authors finally report no alteration of IR-induced effects by CRIPS.

However, there are several major concerns on the context of this manuscript:

  1. It needs to be clearly mentioned in the text that CRIPS and IR factors were applied in a different manner: СRIPS as a chronical exposure, while Fe-IR as an acute irradiation. In this context, the basic conclusion claiming an absence of alteration of IR-induced effects by CRIPS is still far from practical interpretation, at least for deep-space missions, where IR applied chronically as well. A special note in the “Discussion” section needs to be added that possible different pattern of CRIPS and Fe-IR synergy can take place if both these factors are applied chronically. The conclusion made by authors might have a practical implication if an appropriate way to simulate chronical IR-exposure is applied, for instance, fractioned exposures within the period of restraining.

  1. The radiotherapy context mentioned by the authors also does not meet the design of the experiment. Fe irradiation applied in the mentioned doses and within a single fraction is far from practically used carbon, proton, X- or gamma-ray therapy issues.

  1. The context of ground-based nuclear accidents, like in Chernobyl, is also confusing within the current study. First, because types of radiation are different (sparsely ionizing radiation in nuclear accidents vs. densely ionizing Fe ions in current study), second, because of the different nature of exposure (nuclear accidents can be associated with extremely high doses or/and incorporated radionuclides, while space radiation is delivered in relatively low doses and lacks incorporation). This makes expected physiological outcomes drastically different in these two contexts. It is especially important when we speak about the effects in organs like kidneys, which are associated with an overall physiological state of the organism.

  1. Description of the experiment design lacks of the important information on whether irradiation occurred during the restraining session or between them. This may influence on the tissue oxygen saturation and, therefore, determine the level of ROS production and the final irradiation outcome.

  1. First paragraph of the “Discussion” section looks like authors consider CRIPS as a model of the stress caused by scare of a certain disaster. Meanwhile, psychophysiological and mental backgrounds of the restraining and the scare of a certain disaster are different. Especially if we speak about events happened in the past, or persistent stress caused by chronic fear of something (say, of nuclear disaster or any other). Therefore, the stress model used in the study also needs to be placed in a right context.

  1. Generally, the current paper seems to be a part of a series of studies by authors where they sequentially probe a synergy between CRIPS and IR through the multiple molecular and physiological outcomes. However, most of the results indicate absence of any significant influence of CRIPS on the considered radiation-induced effects, except the chromosomal exchanges. Moreover, the data reported by authors in two past pares, namely [1] and [7] does not allow to identify a clear regularity in the observed effect: paper [7] suggests that CRIPS does not appear to synergize with the clastogenicity of IR, except the effect on translocations, which level appeared at a lower frequency in mice exposed to 4 Gy of X-ray-IR plus CRIPS than in those exposed to IR alone; meanwhile, in paper [1] authors found an enhancing effect of CRIPS on chromosomal exchanges induced by Fe-IR at a low dose of 0.1 Gy. Therefore, the overall mission of such a series of studies with lots of results demonstrating no expected effect needs to be clarified and, probably, re-defined.

Author Response

IJMS-1671802: Effects of concurrent exposure to chronic restraint-induced stress and total-body iron ion radiation on induction of kidney injury in mice

 Dear editor, 

Thank you so much for the thoughtful and constructive feedback that you and our reviewers provided regarding our manuscript.We have amended the manuscript thoroughly according to our reviewers’ constructive comments and carefully addressed each of the questions raised by our reviewers.  We are certain that you and our reviewers will find the revised version of our manuscript clears up all the issues that the reviewers kindly indicated. With the critical changes to our manuscript, we hereby resubmit revised version of our manuscript for a secondary evaluation. Thank you once again for your consideration of our paper. 

Sincerely yours, 

Authors

Reviewer #1:

The study deals with probing the effect of concurrent exposure to ionizing radiation (IR) and chronic restraint-induced stress (CRIPS) through the renal damage in Trp53-heterozygous mice.

The methods used in the study are clear and the results are described in detail. As the main conclusion, authors finally report no alteration of IR-induced effects by CRIPS.

However, there are several major concerns on the context of this manuscript:

  1. It needs to be clearly mentioned in the text that CRIPS and IR factors were applied in a different manner: СRIPS as a chronical exposure, while Fe-IR as an acute irradiation. In this context, the basic conclusion claiming an absence of alteration of IR-induced effects by CRIPS is still far from practical interpretation, at least for deep-space missions, where IR applied chronically as well. A special note in the “Discussion” section needs to be added that possible different pattern of CRIPS and Fe-IR synergy can take place if both these factors are applied chronically. The conclusion made by authors might have a practical implication if an appropriate way to simulate chronical IR-exposure is applied, for instance, fractioned exposures within the period of restraining.

We appreciate the valuable and thoughtful comments of the reviewer. We’ve got many ideas for our future work. Following those feedback, we have carefully revised the manuscript accordingly. Please kindly note that some significantly changed parts (track change) in the manuscript.

  1. The radiotherapy context mentioned by the authors also does not meet the design of the experiment. Fe irradiation applied in the mentioned doses and within a single fraction is far from practically used carbon, proton, X- or gamma-ray therapy issues.

 We totally agree with the reviewer’s comments. We have removed the radiotherapy context in the revised manuscript accordingly.

  1. The context of ground-based nuclear accidents, like in Chernobyl, is also confusing within the current study. First, because types of radiation are different (sparsely ionizing radiation in nuclear accidents vs. densely ionizing Fe ions in current study), second, because of the different nature of exposure (nuclear accidents can be associated with extremely high doses or/and incorporated radionuclides, while space radiation is delivered in relatively low doses and lacks incorporation). This makes expected physiological outcomes drastically different in these two contexts. It is especially important when we speak about the effects in organs like kidneys, which are associated with an overall physiological state of the organism.

We totally agree with the reviewer’s comments and we have removed the description of nuclear accidents in the revised manuscript accordingly.

  1. Description of the experiment design lacks of the important information on whether irradiation occurred during the restraining session or between them. This may influence on the tissue oxygen saturation and, therefore, determine the level of ROS production and the final irradiation outcome.

We appreciate the valuable and thoughtful comments of the reviewer. We have added the description of the experiment design that is “In the early morning (3:30–04:30 a.m. or 6:00–7:00 a.m.), on day 8 of the 28-day restraint regimen, 56Fe ions of TBI were irradiated at NIRS with a heavy ion medical accelerator in Chiba (HIMAC) at doses of 0.085 Gy/min, 1.1–2.7 Gy/min and 0.1 and 2 Gy (500 MeV/nucleon, 200 keV/µ m)”

  1. First paragraph of the “Discussion” section looks like authors consider CRIPS as a model of the stress caused by scare of a certain disaster. Meanwhile, psychophysiological and mental backgrounds of the restraining and the scare of a certain disaster are different. Especially if we speak about events happened in the past, or persistent stress caused by chronic fear of something (say, of nuclear disaster or any other). Therefore, the stress model used in the study also needs to be placed in a right context.

We appreciate the valuable and thoughtful comments of the reviewer. The first paragraph was revised.

  1. Generally, the current paper seems to be a part of a series of studies by authors where they sequentially probe a synergy between CRIPS and IR through the multiple molecular and physiological outcomes. However, most of the results indicate absence of any significant influence of CRIPS on the considered radiation-induced effects, except the chromosomal exchanges. Moreover, the data reported by authors in two past pares, namely [1] and [7] does not allow to identify a clear regularity in the observed effect: paper [7] suggests that CRIPS does not appear to synergize with the clastogenicity of IR, except the effect on translocations, which level appeared at a lower frequency in mice exposed to 4 Gy of X-ray-IR plus CRIPS than in those exposed to IR alone; meanwhile, in paper [1] authors found an enhancing effect of CRIPS on chromosomal exchanges induced by Fe-IR at a low dose of 0.1 Gy. Therefore, the overall mission of such a series of studies with lots of results demonstrating no expected effect needs to be clarified and, probably, re-defined.

We thank the reviewer for the constructive comment. The related descriptions were revised to clarify the results obtained in a series of investigations. In brief, enhanced effects were observed using endpoints of spenocyte chromosome aberration and erythrocyte genotoxicity in Trp53-heterous mice exposed to CRIPS and Fe particle radiation while no significant effect from CRIPS on IR-induced testis damage and renal injury was found in Trp53-heterozygous mice. On the other hand, in Trp53 wild-type mice no significant effect from CRIPS on IR-induced chromosome aberration and erythrocyte genotoxicity was observed.

(For your possible reference, the detailed results are summarized as follows:

We used two types of animals (Trp53-heterozygous mice and Trp53 wild-type mice) and two types of radiation (X-rays and Fe particles). For the effects from PS on induction of genotoxicity by Fe-particle radiation, exposure to either CRISP or Fe-particle radiation (0.1 Gy) did not induce a marked increase in the incidence of micronucleated erythrocytes, while concurrent exposure to CRIPS and Fe-particle radiation (0.1 Gy) caused significantly higher increase in the incidence [Katsube et al. Radiat. Res., 2021, 196(1): 100–112.]. Of note, in Trp53 wild-type mice, previous investigations using the same CRIPS model showed that CRIPS alone did not cause increased chromosomal aberrations in splenocytes and elevated micronuclei in bone marrow erythrocytes. CRIPS did not appear to synergize with the TBI (4.0 Gy X-ray radiation) clastogenicity and genotoxicity [Wang et al. J. Radiat. Res., 2015, 56(5): 760–7.]. Moreover, in splenocytes collected from the same Trp53-heterozygous mice, our recent work demonstrated that neither CRIPS nor TBI (0.1 Gy Fe-particle radiation) alone induced any increase in the frequency of aberrant chromosomes while concurrent exposure resulted in a statistically significant increase in the frequency of chromosomal exchanges [Wang et al. Life, 2022, 12, 565.]. In addition, CRIPS has no additive effects on IR-induced renal damage and apoptosis of spermatogenic cells in Trp53-heterozygous mice [Li et al. Int. J. Biol. Sci., 2018, 14(9): 1109–1121.]. These results suggested the functional Trp53 dynamics vary between tissues and are frequently implicated in contributing to radiation sensitivity through activating subsets of target genes to carry out cell fates (i.e., apoptosis, cell cycle arrest, and DNA repair). Together, these findings also suggest that studies on concurrent exposure to PS and IR should be carried out using different endpoints in different tissues and in animals with different genetic background.)

Reviewer 2 Report

“Effects of concurrent exposure to chronic restraint-induced stress and total-body iron ion radiation on induction of kidney injury in mice” by Li et al. studied in vivo the effects of concurrent ionizing radiation exposure and psychological stress on renal damage.

The article is interesting, with promising results and observations, but there are some aspects that requires improvement. Please take into consideration the following remarks:

  1. You should clearly highlight in the introduction, the novelty of the study.
  1. I suggest you to add the details related to the 6 mice groups in a table, to be easy for the readers to follow this important data (lines 93-112). Have you used a software to calculate the number of animals included in your study?
  2. Line 136 - 3,3’-diaminobenzidine, has a different style.
  3. Please provide full information about used reagents and instruments (company, city, country): line 123 – what kind of buffer, line 124 – glacial acetic acid, line 134 – HRP, TBST, line 136 – DAB, and so on.
  4. Please use the same line spacing, suggested by the manuscript template (figures titles, lines 219-227).
  5. In Discussion section, some sentences are simply enumerated, without being connected.
  6. Even if Conclusion section is not mandatory, I suggest you to add one.

Author Response

IJMS-1671802: Effects of concurrent exposure to chronic restraint-induced stress and total-body iron ion radiation on induction of kidney injury in mice 

Dear editor, 

Thank you so much for the thoughtful and constructive feedback that you and our reviewers provided regarding our manuscript.We have amended the manuscript thoroughly according to our reviewers’ constructive comments and carefully addressed each of the questions raised by our reviewers.  We are certain that you and our reviewers will find the revised version of our manuscript clears up all the issues that the reviewers kindly indicated. With the critical changes to our manuscript, we hereby resubmit revised version of our manuscript for a secondary evaluation. Thank you once again for your consideration of our paper. 

Sincerely yours, 

Authors

Reviewer #2:

“Effects of concurrent exposure to chronic restraint-induced stress and total-body iron ion radiation on induction of kidney injury in mice” by Li et al. studied in vivo the effects of concurrent ionizing radiation exposure and psychological stress on renal damage.

The article is interesting, with promising results and observations, but there are some aspects that requires improvement. Please take into consideration the following remarks:

1. You should clearly highlight in the introduction, the novelty of the study.

We appreciate the valuable and thoughtful comments of the reviewer. We have revised the introduction and highlighted the novelty of this study in the first and second paragraph.

2.  I suggest you to add the details related to the 6 mice groups in a table, to be easy for the readers to follow this important data (lines 93-112). Have you used a software to calculate the number of animals included in your study?

We thank the reviewer for the constructive comment. We have added the details related to the 6 mice groups in the Table 1. In fact, more than 6 mice were used in each of the experimental groups, and for the present work on renal injury, kidney samples from 6 mice were analyzed. Using at least 6 samples in each group was recommended by our institutional statistician.

3. Line 136 - 3,3’-diaminobenzidine, has a different style.

We appreciate the thoughtful comments of the reviewer. The same style of 3,3’-diaminobenzidine was used in the revised manuscript.

4. Please provide full information about used reagents and instruments (company, city, country): line 123 – what kind of buffer, line 124 – glacial acetic acid, line 134 – HRP, TBST, line 136 – DAB, and so on.

We appreciate the thoughtful comments of the reviewer. We have added the full information of reagents and instruments when first appeared in the text.

5. Please use the same line spacing, suggested by the manuscript template (figures titles, lines 219-227).

We appreciate the thoughtful comments of the reviewer. We have used the same line spacing in the revised manuscript.

6. In Discussion section, some sentences are simply enumerated, without being connected.

We appreciate the thoughtful comments of the reviewer. We have revised the “Discussion” and the sentences were connected fluently.

7. Even if Conclusion section is not mandatory, I suggest you to add one.

We thank the reviewer for the constructive comment. We have separated the summary as the “Conclusion” section in the revised manuscript.